

# Identification of key genes in bovine muscle development by co-expression analysis

Junxing Zhang[1,*], Hui Sheng[1,*], Cuili Pan[1], Shuzhe Wang[1], Mengli Yang[1], Chunli Hu[1], Dawei Wei[1], Yachun Wang[2] and Yun Ma[1]

[1] Ningxia University, Yinchuan, China
[2] China Agricultural University, Beijing, China
[*] These authors contributed equally to this work.

## ABSTRACT

**Background**. Skeletal muscle is not only an important tissue involved in exercise and metabolism, but also an important part of livestock and poultry meat products. Its growth and development determines the output and quality of meat to a certain extent, and has an important impact on the economic benefits of animal husbandry. Skeletal muscle development is a complex regulatory network process, and its molecular mechanism needs to be further studied.

**Method**. We used a weighted co-expression network (WGCNA) and single gene set enrichment analysis (GSEA) to study the RNA-seq data set of bovine tissue differential expression analysis, and the core genes and functional enrichment pathways closely related to muscle tissue development were screened. Finally, the accuracy of the analysis results was verified by tissue expression profile detection and bovine skeletal muscle satellite cell differentiation model *in vitro* (BSMSCs).

**Results**. In this study, *Atp2a1*, *Tmod4*, *Lmod3*, *Ryr1* and *Mybpc2* were identified as marker genes in muscle tissue, which are mainly involved in glycolysis/gluconeogenesis, AMPK pathway and insulin pathway. The assay results showed that these five genes were highly expressed in muscle tissue and positively correlated with the differentiation of bovine BSMSCs.

**Conclusions**. In this study, several muscle tissue characteristic genes were excavated, which may play an important role in muscle development and provide new insights for bovine molecular genetic breeding.

## INTRODUCTION

The skeletal muscle of livestock and poultry is the main source of meat products, providing human beings with high-quality animal protein and nutrients. With the booming of the livestock industry, the muscle yield and quality of livestock and poultry are gaining more and more attention and importance. However, the regulatory process of skeletal muscle growth and development involves the activation or silencing of numerous genes and their associated pathways, and is an extremely complex multi-level regulatory network.

Corresponding authors
Yachun Wang, wangyachun@cau.edu.cn
Yun Ma, mayun_666@126.com

Therefore, revealing the process of skeletal muscle development and its regulatory factors will help to provide ideas for the improvement of livestock and poultry meat quality.

At present, a large number of public transcriptome data appear with the development of high-throughput sequencing technology, which provides valuable resources for screening biomarkers related to livestock and poultry development (*Rao et al., 2021*). The weighted co-expression network (WGCNA) is a systems biology method that can analyze microarray data to explore the correlation between genes (*Langfelder & Horvath, 2008*). It can distinguish the associations between different gene sets and between trait phenotypes and gene sets, and gather highly related genes into the same module (*Sheng et al., 2022*). The genes in the module can be organized by connectivity. Gene-to-gene interactions are revealed at the systematic level, which will help researchers to further understand the mechanisms behind gene interactions and to identify candidate biomarkers or therapeutic targets (*Talukdar et al., 2016*). In fact, the use of WGCNA has played an important role in revealing the effects of meat traits and related metabolism (*Ponsuksili et al., 2012*), as well as genetic variation in the study of cellular functions and diseases (*Farber & Lusis, 2009*; *Calabrese et al., 2012*).

In this study, we used WGCNA to analyze the GSE137943 dataset and successfully screened five hub genes associated with bovine muscle tissue by constructing gene co-expression networks from transcript data of multiple tissues and analyzing the main functions of trait-related module genes. Finally, we used tissue samples and bovine BSMSCs to detect the expression pattern of hub gene, preliminarily analyzed the function of hub gene, and verified the reliability of the analysis results. We hope that the results of this study will contribute to the study of the regulatory mechanism of bovine muscle development.

## MATERIAL AND METHODS

### Ethics statement

Animal experiments were conducted according to the guidelines of the Regulations for the Administration of Affairs Concerning Experimental Animals (Ministry of Science and Technology, China, 2004). With the approval of the Animal Welfare Committee of Ningxia University (license no. NXUC20211058), we first injected a premature calf with a sedative, and when the calf entered a quiet state, we injected an excessive amunt of barbiturate (100 mg/kg) intravenously, making every effort to relieve the pain of the animal. Then the muscle tissue of calves was collected and the primary bovine BSMSCs was isolated, cultured and differentiated immediately. meanwhile, tissue samples of kidney, liver, heart, muscle, rumen, adipose, spleen and lung of calves were collected, and these tissue samples were stored in liquid nitrogen for subsequent experimental verification. In addition, the pregnant cow was not sampled, and after a period of rest and recuperation, it was still raised in the Fumin farm in Guyuan (Yinchuan, China).

### Data collection

GSE137943 transcriptome data were mainly used to analyze the differences of gene expression among different tissues of cattle, which were provided by GEO database

(https://www.ncbi.nlm.nih.gov/geoprofiles) (*Fang et al., 2020*). We analyzed the expression profile data of 37 samples from eight tissues (adipose, heart, kidney, muscle, liver, lung, rumen and spleen) of cattle in this data set. All RNA-seq data sets are preprocessed using the robust multichip average (RMA) for background correction and normalization (*Irizarry et al., 2003*).

## Construction of co-expression network

The gene co-expression network was constructed by using WGCNA software package in R (*Pan et al., 2022*). First, a similarity matrix was constructed by calculating Pearson correlation coefficients. Then, a scale-free topological network is constructed by using PickSoftThreshold function and selecting appropriate soft threshold parameter $\beta$ (*Chen et al., 2018*). The adjacency matrix is then transformed into a topological overlap matrix (TOM), which is capable of computing the network connectivity of genes for network generation (*Botia et al., 2017*). We chose the soft threshold power (8) corresponding to a correlation coefficient threshold of 0.8 to construct a scale-free co-expression network, defined 0.25 as the threshold for dividing the modules, and set the minimum number of genes in the modules to 30.

## Identify module of interest and functional annotation

The WGCNA algorithm took the two significant parameters module eigengene (ME) and module significance (MS) to evaluate the modules related to muscle tissue. ME is defined as the main parameter of principal component analysis, which can transform the expression of genes in specific modules into characteristic expression profiles (*David & Jacobs, 2014*). MS was defined as the mean value of gene significance (GS) in the module, where the module with the highest absolute value of MS was considered to be the most associated with muscle tissue. Functional enrichment analysis of genes in muscle tissue related modules was performed using the clusterprofiler package in R (*Yu et al., 2012*). When the $P < 0.01$, it was considered significantly enriched in GO-BP process and KEGG pathway.

## Hub genes identification

Module members (MM) are mainly used to analyze the correlation between gene expression profile and ME, and gene significance (GS) can reflect the correlation between phenotype and gene expression value (*Liu et al., 2019*). We used GS $\geq$ 0.79 and MM $\geq$ 0.80 as criteria to screen candidate hub genes in the co-expression network. The STRING website (https://string-db.org/) was used to construct a network of PPI interactions associated with muscle tissue, and the CytoHubba plug-in for Cytoscape software was used to identify central genes in the PPI network (*Chin et al., 2014*; *Szklarczyk et al., 2017*). Finally, the common genes identified by co-expression network and PPI network are defined as "real" Hub genes (*Li, Pu & Wu, 2019*; *Lu et al., 2020*).

## Single-gene gene set enrichment analysis

Gene set enrichment analysis (GSEA) is a computational method that analyzes the form of gene expression in a specific functional gene set and whether this expression form is

statistically significant (*Subramanian et al., 2005*). We used the GSEABase package in R to analyze the expression matrix of the samples, using intergenic correlation as a criterion for grouping (*Merico et al., 2010*).

## Cell isolation and culture

According to the results of previous studies, bovine hind limb muscle tissue was collected, cut into small pieces and bovine BSMSCs were isolated using the type II collagenase (Gibco, USA) and trypsin (Gibco, Waltham, MA, USA) method (*Dai et al., 2016*). Then, the isolated cells were inoculated into a cell petri dish containing proliferation medium (20% FBS + 80% DMEM) for culture. Until the confluence approximately reaches 70%, the differentiation medium (2% FBS + 98% DMEM) was replaced to induce cell differentiation (*Zhang et al., 2020*).

## RNA isolation and qRT-PCR

Tissue samples of newborn calves and 2.5-year-old calves (heart, liver, spleen, lung, kidney, muscle, fat and rumen epithelium) were provided by the Key Laboratory of Ruminant Molecular Cell breeding of Ningxia University. Trizol reagent (Invitrogen, Waltham, MA, USA) was used to extract the total RNA of bovine BSMSCs from different tissue samples and different culture periods (GM: proliferation phase; DM1~5: 1–5 days of differentiation). Total RNA was reverse transcribed using the PrimeScript II 1st Strand cDNA Synthesis Kit (Takara, Dalian, China), and real-time quantitative real-time PCR (qRT-PCR) experiments using reverse transcribed products were performed to measure the mRNA expression levels of target genes (*Chen et al., 2021*). The primer information is shown in Table S8. The qRT-PCR program is added in the Table S9.

## Immunofluorescence assay

After washing the cells with PBS, the cells were first fixed with 4% paraformaldehyde for 30 min, then treated with PBS containing 0.1%TritonX-100, then blocked with 5% bovine serum albumin (BSA) at room temperature for 30 min, and finally diluted anti-MyHC antibody (1:100, Abcam, Cambridge, MA, USA) was added overnight treatment at 4 °C. On the second day, the cells were washed with PBS for 3 times and incubated with FITC-conjugated goat anti-rabbit IgG-labeled secondary antibody for 1 h at room temperature and in the absence of light. The cells were washed with PBS three times, and then stained for 5 min with 4,6-diamino-2-phenylindole (DAPI). Finally, the cells were observed and photographed under an inverted fluorescence microscope.

## Expression analysis of hub genes

The differential expression level of hub gene in GSE137943 data set was analyzed, and the analysis results were verified by GSE116775 data set (*Sun et al., 2021*). In addition, the mRNA level of hub gene in bovine BSMSCs from different tissue samples and different culture periods was also detected.

## Statistical analysis

All experimental results were calculated as mean ± SEM of three independent experiments. Double-tailed $t$-test or chi-square test was used to compare between groups, and the

expression level of GAPDH was used as endogenous control, and was considered statistically significant when the $P < 0.05$ (*$P < 0.05$ and **$P < 0.01$).

## RESULTS

### Construction of co-expression network

In this study, WGCNA analysis was performed based on expression profiles from the GSE137943 dataset (Table S1) to detect the gene clusters most relevant to muscle tissue. The results of sample cluster analysis show that the same tissue samples will be divided into the same cluster, indicating that the tissue type is the main reason for the differences between samples (Fig. 1). In this study, the soft threshold parameters corresponding to a correlation coefficient threshold of 0.8 were chosen to ensure a scale-free network (Fig. 2A). In the GSE137943 dataset, 23 co-expression modules were constructed, and the module with the most genes was turquoise module (982), followed by yellow module (854) and green module (622) (Fig. 2B). The characteristic neighbourhood heat map represented in Fig. 2C shows the relationships between genes in different modules in the dataset, with genes in the same module being strongly correlated with each other, while genes in different modules are almost independent of each other.

### Analysis of muscle tissue related modules

We obtain the module-feature heat map by calculating the correlation coefficient between the module and the feature of the data set. According to the GS algorithm, we found that the correlation between magenta module and muscle tissue ($r = 0.93$, p $= 8E^{-11}$) in GSE137943 data set was the most significant (Fig. 3A), and there were 223 genes in magenta module (Table S3). Analysis of the genes in the magenta module is important for the study of bovine muscle tissue and the highest expression levels of genes associated with muscle tissue were found in this module (Figs. 3A, 3B). Bioinformatics studies have shown that these muscle tissue-associated genes are enriched in biological processes such as muscle contraction and muscle structure development, and are involved in signaling pathways such as cGMP-PKG and cAMP signaling pathways (Fig. 4). All enrichment terms and explanations can be found in Table S4.

### Excavation of the hub gene

We screened 20 co-expressed hub genes in the magenta module using GS $\geq 0.79$ and MM $\geq 0.80$ as criteria, and these genes have close interrelationships with each other (Fig. 5A; Table S5). Meanwhile, the PPI network was constructed for the genes in the magenta module, and the 20 genes with the highest connectivity in the network were selected (Fig. 5B; Table S6). Finally, the intersection was taken for the 40 genes and five common genes (*Tmod4*, *Ryr1*, *Mybpc2*, *Lmod3* and *Atp2a1*) were defined as the "real" hub genes (Fig. 5C).

### Gene set enrichment analysis

To analyze the potential functions of *Atp2a1*, *Tmod4*, *Lmod3*, *Mybpc2* and *Ryr1* genes, we performed GSEA analysis. The results showed that the genes positively correlated with hub gene in GSE137943 dataset were mainly concentrated in insulin signal pathway, calcium

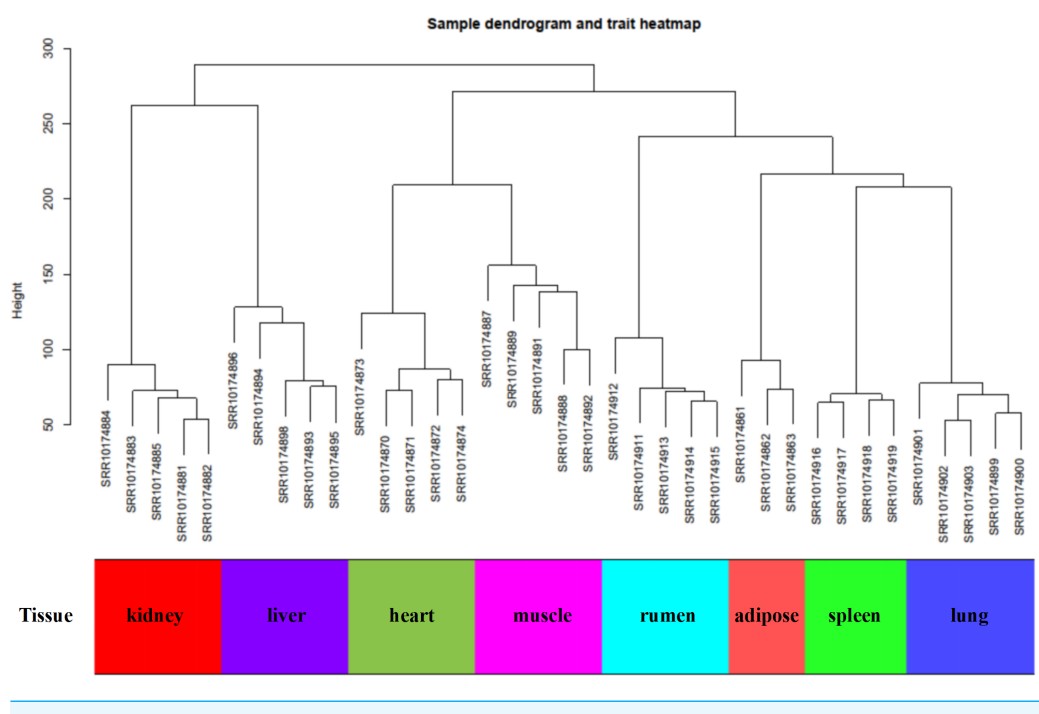

**Figure 1** Sample dendrogram of GSE137943 dataset.

signal pathway, AMPK signal pathway, MAPK signal pathway, oxidative phosphorylation and glycolysis/gluconeogenesis (Fig. 6; Table S7).

## Validation of the hub genes

We analyzed the differential expression levels of hub genes in GSE137943 data sets and found that these genes were highly expressed in muscle tissue (Fig. 7). Meanwhile, the expression levels of the five hub genes mentioned above were examined using the GSE116775 dataset (Table S2). As shown in Fig. 8, in the GSE116775 data set, the expression levels of *Atp2a1* (Fig. 8A), *Tmod4* (Fig. 8B), *Lmod3* (Fig. 8C), *Mybpc2* (Fig. 8D) and *Ryr1* (Fig. 8E) in muscle tissue were significantly higher than those in other tissues. In addition, we used tissue samples from newborn calves and adult cattle collected in the laboratory to detect the expression level of hub gene, and found that the results were consistent with the expression trend of GSE137943 data set and GSE116775 data set (Figs. 9 and 10).

## Bovine BSMSCs differentiation culture

Myosin heavy chain (MyHC) and Myogenin (MyoG) are the differentiation markers of bovine BSMSCs. We used real-time fluorescence quantitative PCR reaction to detect the expression level of marker genes during cell differentiation. The results showed that the mRNA expression levels of *MyoG* and *MyHC* in differentiated bovine BSMSCs were significantly higher than those before differentiation. Meanwhile, immunofluorescence staining analysis of bovine BSMSCs from GM, DM3 and DM5 was carried out. The results showed that no myotubes were observed before the differentiation of bovine BSMSCs, and the fluorescence signal intensity of MyHC was weak. However, after induced

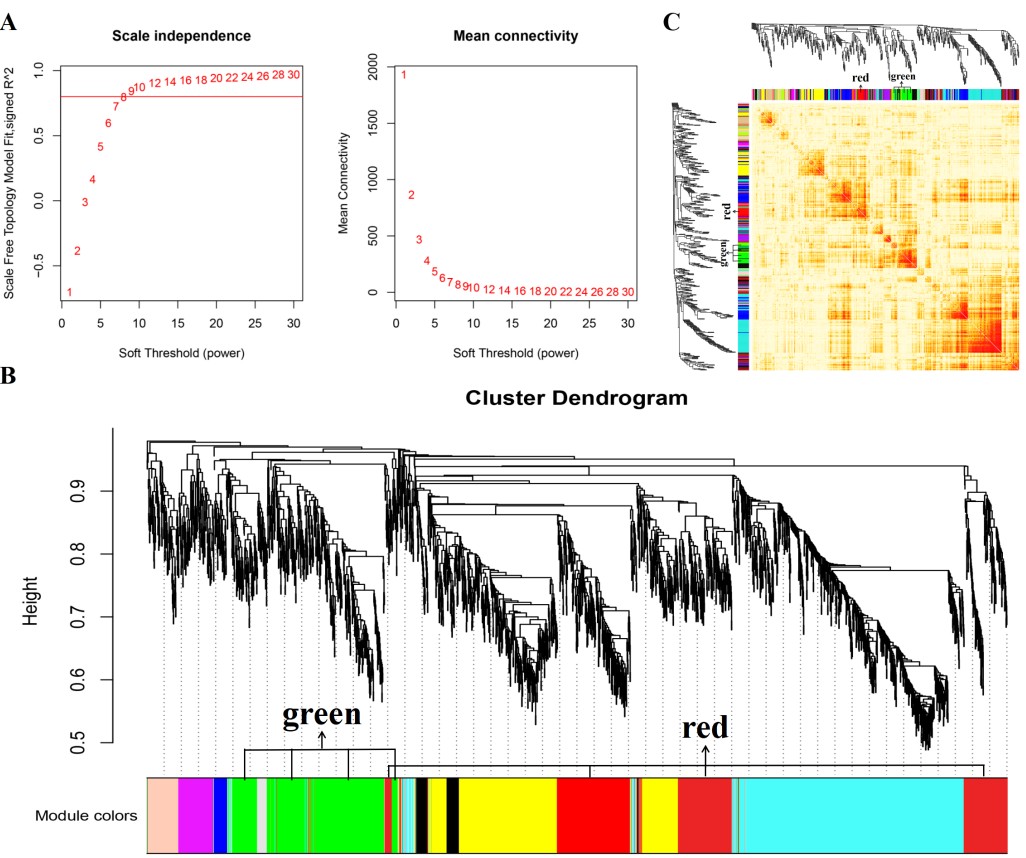

**Figure 2** **Gene co expression network was constructed by GSE137943 dataset.** Analysis of the scale-free fit index for soft-thresholding powers (left) and the mean connectivity for various soft-thresholding powers (right); (B) Gene expression clustering tree and recognition module in co-expression network; (C) network heatmap plot in the co-expression modules.

differentiation, bovine BSMSCs fused and formed muscle tubes, and the expression level of MyHC increased significantly (Fig. 11). The above results indicate that a model of induced differentiation of bovine BSMSCs was successfully constructed.

### Detection of hub gene expression in Bovine BSMSCs

QRT-PCR was used to detect the expression changes of hub gene during the induction and differentiation of bovine BSMSCs. The results showed that the expression level of hub gene in DM phase cells was significantly higher than that in GM phase cells, indicating that the identified hub gene played a role in myogenic differentiation of bovine BSMSCs, and they played an important role in the study of the molecular mechanism of muscle development (Fig. 12).

## DISCUSSION

The growth and development of skeletal muscle is of great significance to improve meat production performance and product quality. In addition, the analysis of the structural

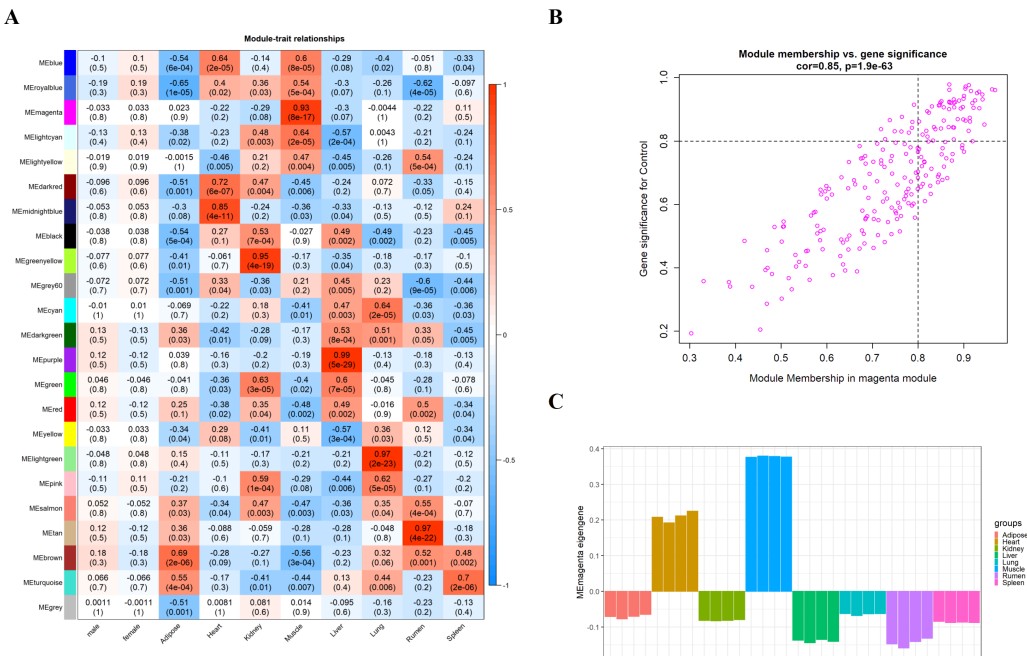

**Figure 3  Module-trait correlations analysis in muscle tissue (GSE137943).** (A) Heat map of correlation between GSE137943 data set module and muscle tissue; (B) Significance of genes related to muscle tissue in the magenta module (each dot represents a gene in the magenta module); (C) Module eigengene ($y$-axis) across samples ($x$-axis) from the magenta module (associated to muscle tissue).

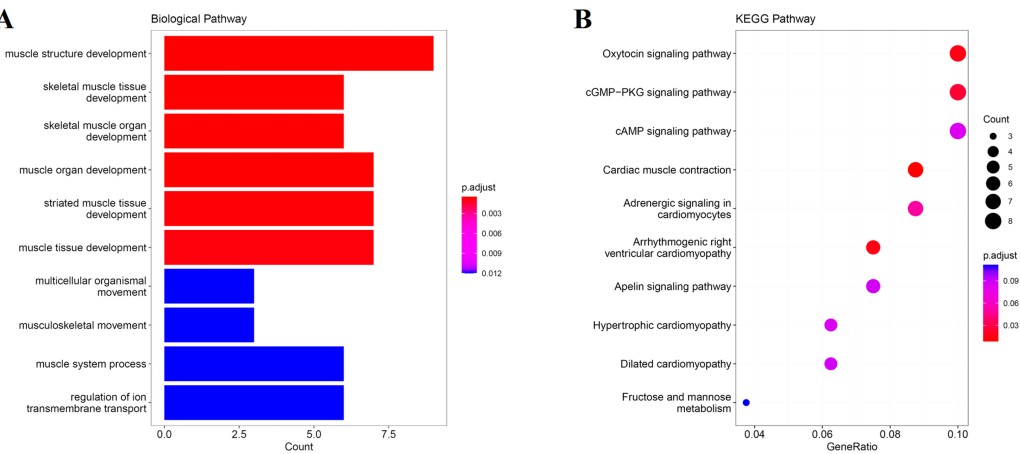

**Figure 4  GO and KEGG analysis.** The visualization results of (A) partial GO biological function analysis and (B) partial KEGG analysis of magenta module gene. The first 10 important enrichment pathways are shown.

characteristics and growth rules of bovine skeletal muscle and the factors regulating the growth and development of bovine skeletal muscle tissue have important implications for beef cattle breeding and production. Therefore, to better understand the genetic and molecular influences behind economic traits of muscle tissue, we used RNA sequencing
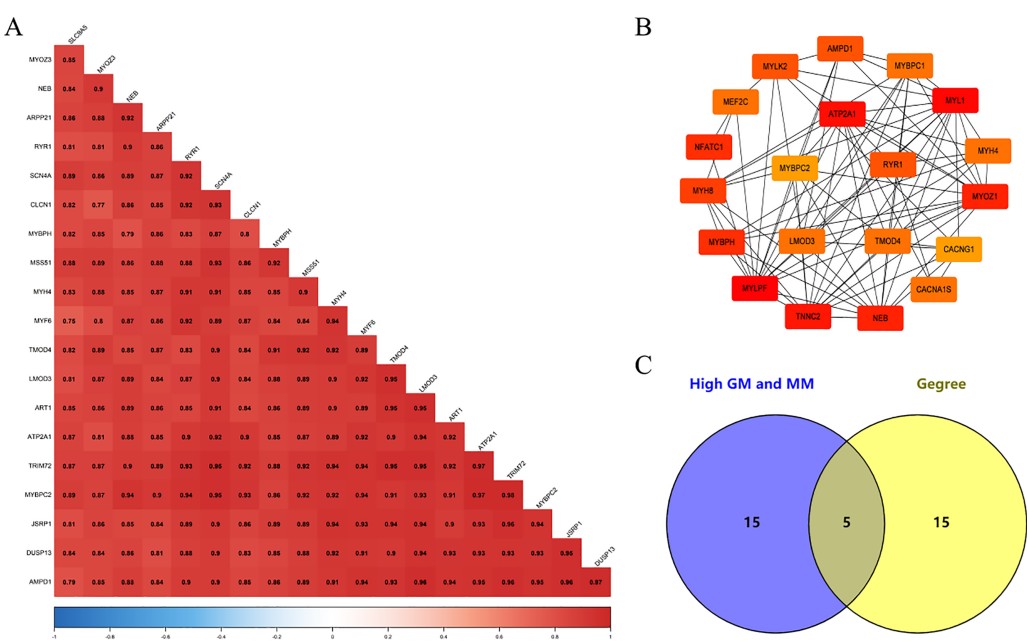

**Figure 5** **Identification of Hub gene.** (A) Correlation of the top 20 genes with high MM and GS in the magenta module; (B) the top 20 genes with the highest connection degree in magenta module were identified by Cytoscape software; (C) identify the common genes in the co-expression network and PPI network.

data and WGCNA tools to construct co-expressed gene networks and identify possible candidate genes and metabolic pathways regulating muscle tissue traits.

Because WGCNA analysis focuses on mining the relationship between target features and co-expression modules, the results have good reliability and biological significance (*Chou et al., 2014*). Therefore, we analysed 37 samples from the GSE137943 dataset using WGCNA and found that the magenta module was significantly associated with muscle tissue (Fig. 3A). The KEGG analysis of genes in magenta module showed that these genes were mainly enriched in cGMP-PKG, cAMP, Calcium signaling pathway and other signal pathways (Fig. 4A), and were closely related to biological processes such as muscle system process, muscle structure development, skeletal muscle tissue development and muscle contraction (Fig. 4B). Then, five real hub genes were identified by intersecting the hub genes identified in the GSE137943 data set co-expression network and the PPI network (Fig. 5). GSEA of hub genes showed that genes positively associated with hub genes were mainly involved in insulin, calcium, AMPK, MAPK, oxidative phosphorylation and glycolysis/gluconeogenesis pathways (Fig. 6). Studies have show that when the body takes in energy through diet, the pancreas $\beta$ cells synthesize and secrete insulin (*Sharma, Garber & Farmer, 2008*). Insulin is essential for maintaining the body's glucose homeostasis and acts primarily by inhibiting hepatic glucose production and stimulating glucose uptake by muscle and fat (*Zhang & Liu, 2014*). Skeletal muscle is the main target tissue of insulin action, and 75% of insulin-mediated glucose uptake occurs in skeletal muscle. Glucose is transported to muscle cells and converted into glycogen. Glycogen synthesis depends

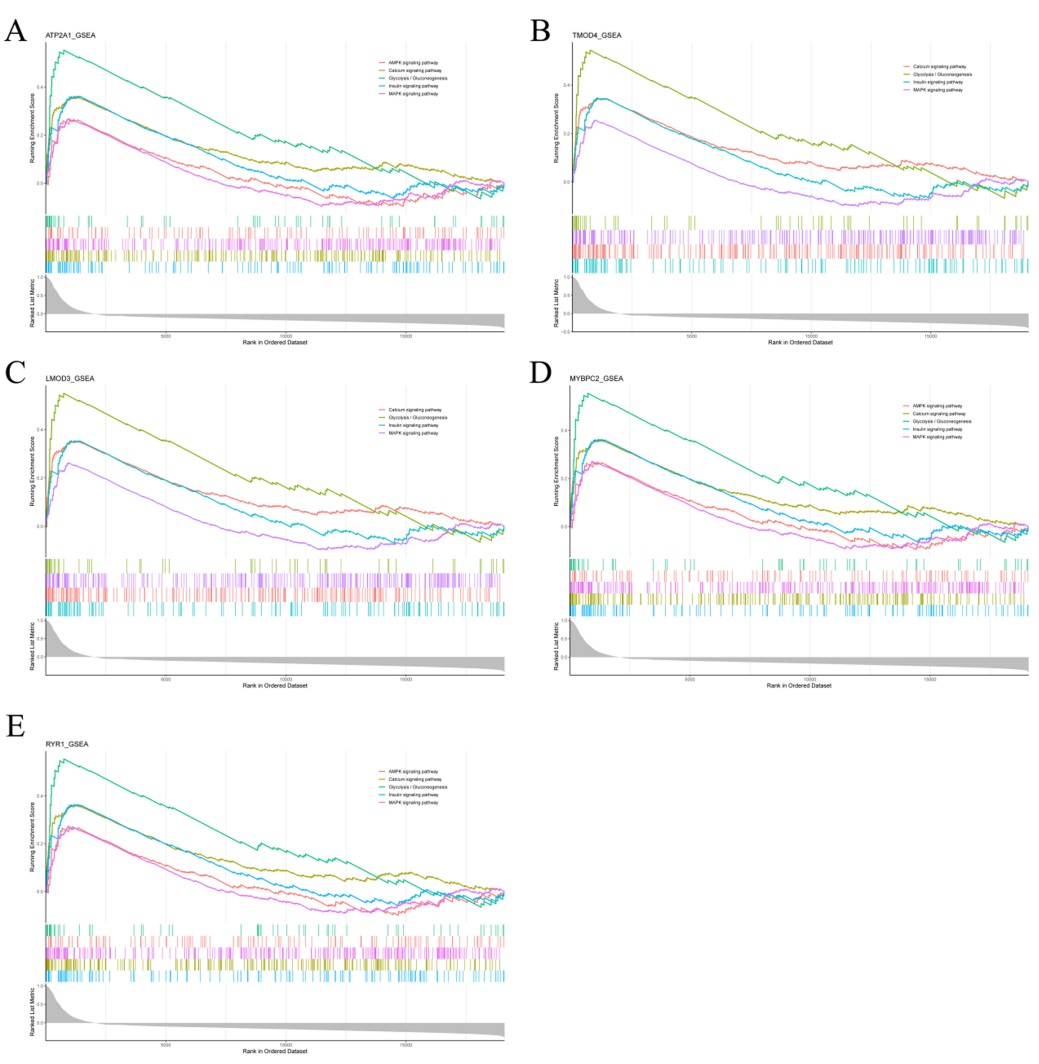

**Figure 6** **The gene set enrichment analysis results of hub gene.** Pathway enrichment analysis of genes positively associated with (A) *Atp2a1*, (B) *Tmod4*, (C) *Lmod3*, (D) *Mybpc2* and (E) *Ryr1* in the GSE137943 dataset.

on glycogen synthase activity and glucose uptake, and insulin signal pathway can affect glycogen synthase activity by regulating GSK-3 $\beta$. Therefore, insulin signaling pathway plays an important role in stabilizing the physiological state of skeletal muscle and maintaining energy balance (*Kampmann et al., 2011*). Calcium signaling pathway can regulate muscle contraction, exocytosis, cell division to gene expression and other cellular functions, which is of great significance for skeletal muscle development, dynamic balance and regeneration (*Tu et al., 2016*). AMPK signal can fuse with PI3K and ERK signals in growth regulation, and with insulin and cAMP dependent pathways in metabolism. It is the main coordinator of cell growth, metabolism and final cell fate (*Mihaylova & Shaw, 2011*). MAPK signal pathway is a kind of phosphorylated signal pathway, which can regulate cell processes such as cell division, differentiation and the release of inflammatory mediators (*Davis,*

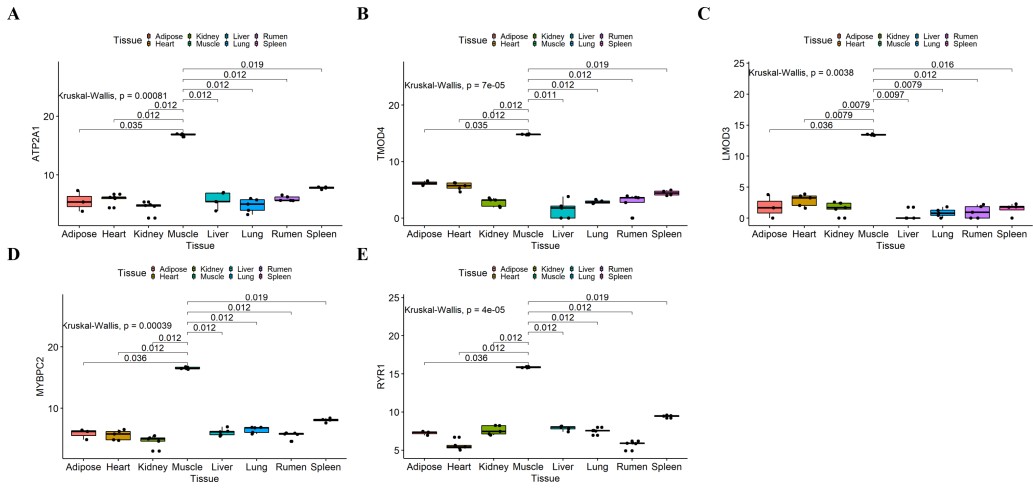

**Figure 7  Expression of hub genes in dataset GSE137943.** (A–E) Expression levels of (A) *Atp2a1*, (B) *Tmod4*, (C) *Lmod3*, (D) *Mybpc2*, and (E) *Ryr1* were significantly increased in muscle tissue.

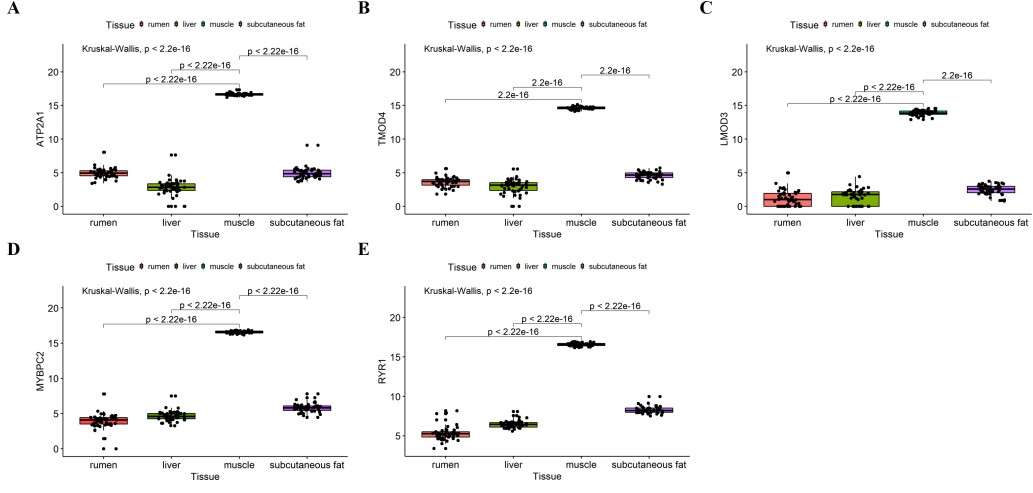

**Figure 8  Expression of hub genes in dataset GSE116775.** (A–E) Expression levels of (A) *Atp2a1*, (B) *Tmod4*, (C) *Lmod3*, (D) *Mybpc2*, and (E) *Ryr1* were significantly increased in muscle tissue.

*2000*; *Wagner & Nebreda, 2009*). Both glycolysis and oxidative phosphorylation play an important role in steady-state ATP synthesis, providing energy for muscle contraction and movement (*Conley, Kemper & Crowther, 2001*). In addition, the hub gene *Atp2a1* is mainly involved in pathways such as muscle contraction and calcium ion transmembrane transport (Fig. 13).

With regard to the identified hub genes, the function of *Atp2a1*, *Tmod4*, *Lmod3*, *Ryr1* and *Mybpc2* in muscle development have been reported. Calcium is the second messenger necessary for cell growth and development, maintenance of intracellular homeostasis and muscle cells to perform special functions (*Lechleiter, John & Camacho, 1998*;

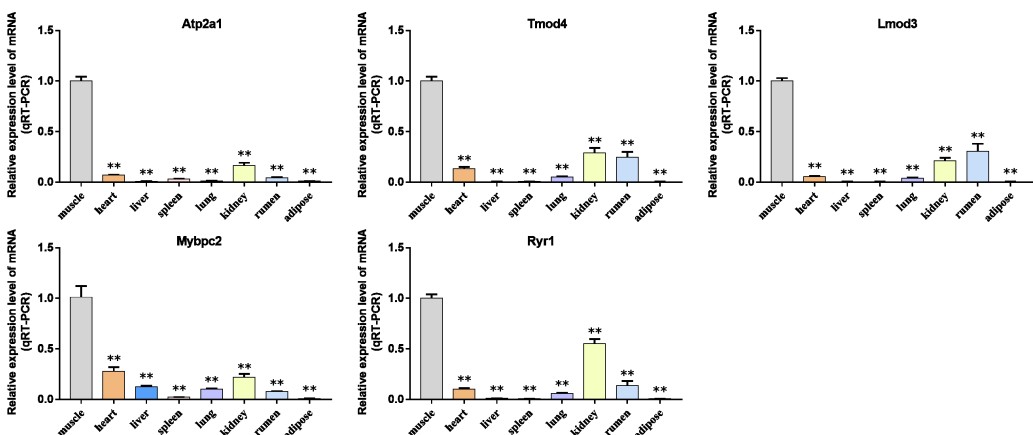

**Figure 9 The expression level of hub gene in newborn calf tissue samples was detected.** (A–E) Compared with other tissues, the expression levels of (A) *Atp2a1*, (B) *Tmod4*, (C) *Lmod3*, (D) *Mybpc2*, and (E) *Ryr1* in muscle tissue were significantly increased.

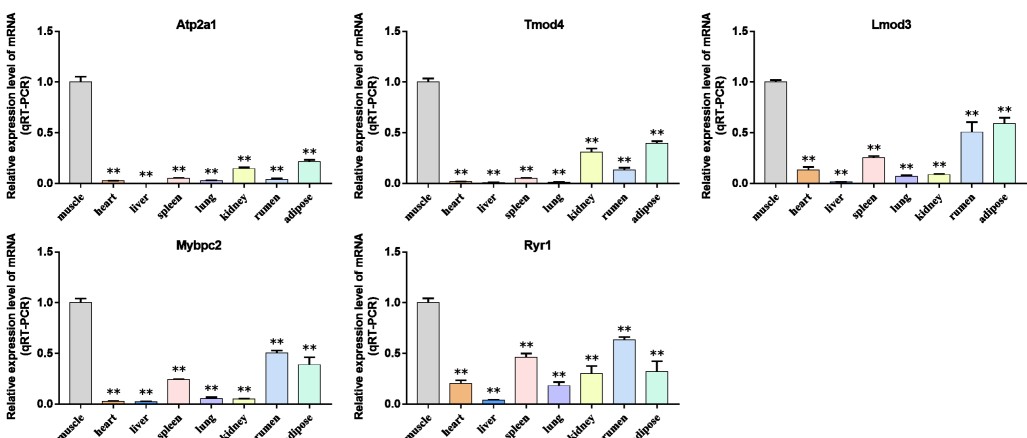

**Figure 10 Expression levels of hub genes were examined in tissue samples from 2.5 year old cattle.** (A–E) Compared with other tissues, the expression levels of (A) *Atp2a1*, (B) *Tmod4*, (C) *Lmod3*, (D) *Mybpc2*, and (E) *Ryr1* in muscle tissue were significantly increased.

*Lipskaia, Hulot & Lompre, 2009*). The sarcoplasmic/endoplasmic reticulum $Ca^{2+}$ ATPases (ATP2As/SERCAs) is the main $Ca^{2+}$ pump of myotube and young muscle fibers, which reduces the level of intracellular $Ca^{2+}$ by accumulating $Ca^{2+}$ into sarcoplasmic reticulum. In human related studies, Atp2a1/Serca1 was found to be highly expressed in fast contracting skeletal muscle; Atp2a2/Serca2 in slow contracting skeletal muscle, vascular myocytes and cardiac myocytes; while Atp2a3/Serca3 was expressed in platelets, lymphocytes, fibroblasts, epithelial cells and endothelial cells (*Periasamy & Kalyanasundaram, 2007*; *Lipskaia, Hulot & Lompre, 2009*). In addition, the study on mouse C2C12 myoblasts showed that the expression of Atp2a1/Serca1b was closely related to the expression of Stim1, Csq and calcineurin, which was necessary for myoblast proliferation and secondary myotube

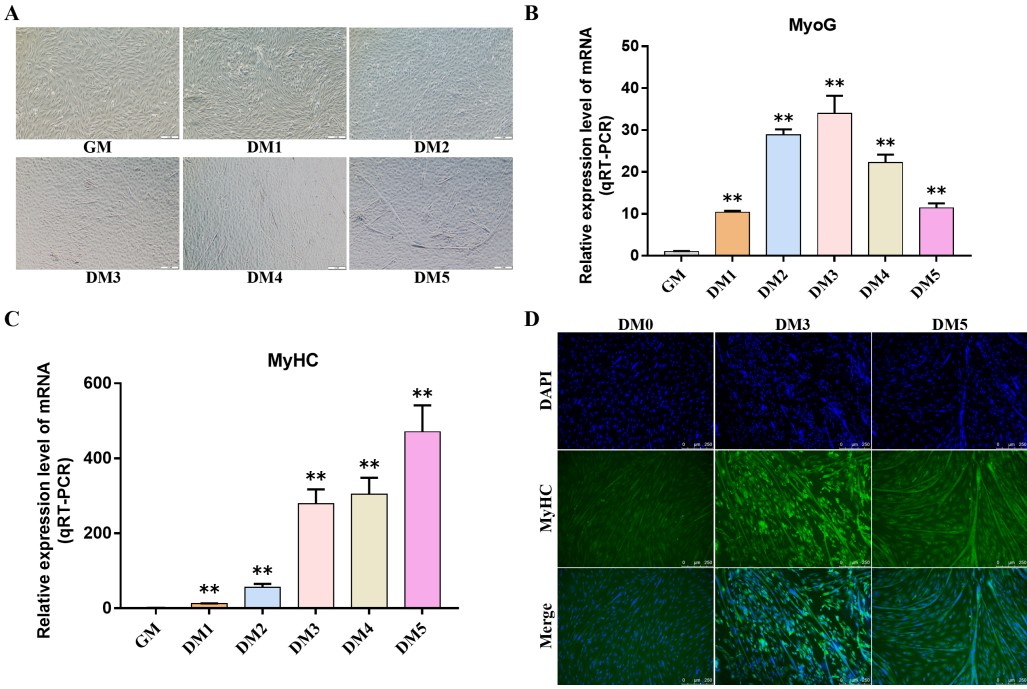

**Figure 11  To induce myogenic differentiation of bovine skeletal muscle satellite cells.** (A) Cell state map of bovine BSMSCs in different culture periods; GM: Proliferative phase, DM1-5: Cell differentiation day 1 to day 5; (B–C) The mRNA expression levels of *MyOG* and *MyHC* in different culture periods were detected; (D) The bovine BSMSCs differentiated for 0 (D0), 3 (D3) and 5 days (D5) were analyzed by immunofluorescence staining (x200). Compared with the control, two asterisks (**) means extremely significant difference (*P* < 0.01).

formation (*Toth et al., 2015*). *Tmod4*, a member of the Tmod family, is highly expressed in skeletal muscle and adipose tissue. Genome-wide comparison of copy number variation between Wannan black pigs (WBP) and Wannan black pigs (AWB) was carried out using genome-wide resequencing data. The results showed that genes such as *Ndn*, *Tmod3*, *Sfrp1* and *Smyd3* were related to muscle development (*Zhang et al., 2022*). It can up-regulate the expression of adipogenic factors, moderately delay muscle development, promote adipogenesis, and play an important role in microfilament length regulation and myofibril assembly (*Zhao et al., 2013*). The functional study of *Lmod3* showed that overexpression of *Lmod3* in C2C12 cells could activate AKT and ERK signaling pathways and promote myoblast proliferation and differentiation, while knocking down the expression of *Lmod3* gene would inhibit cell proliferation and promote cell apoptosis (*Lin et al., 2020*). In addition, it was found that Lmod1-3 is structurally related to Tmod1-4, and these two genes are also located at the tip of actin filaments. The loss of expression of *Tmod4* or *Lmod3* will lead to serious damage to sarcomere assembly and damage to embryonic movement, indicating that appropriate levels of *Tmod4* and *Lmod3* are needed for embryonic muscle fiber formation (*Nworu et al., 2015*). In the skeletal muscle study of mammals from birth to adulthood, it was found that the expression of Ryr1 was always at a high level, and it was found that the relative protein content of Ryr1 in the hindlimb muscle of adult

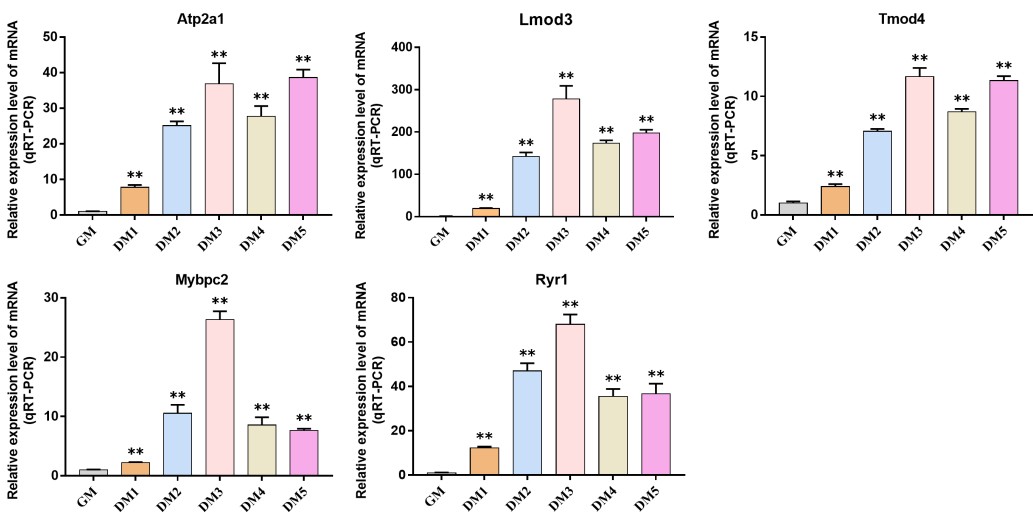

**Figure 12** **Expression levels of hub genes in bovine skeletal muscle satellite cells at different culture periods.** (A–E) Compared with that before differentiation, the expression levels of (A) *Atp2a1*, (B) *Tmod4*, (C) *Lmod3*, (D) *Mybpc2*, and (E) *Ryr1* were significantly increased after induction.

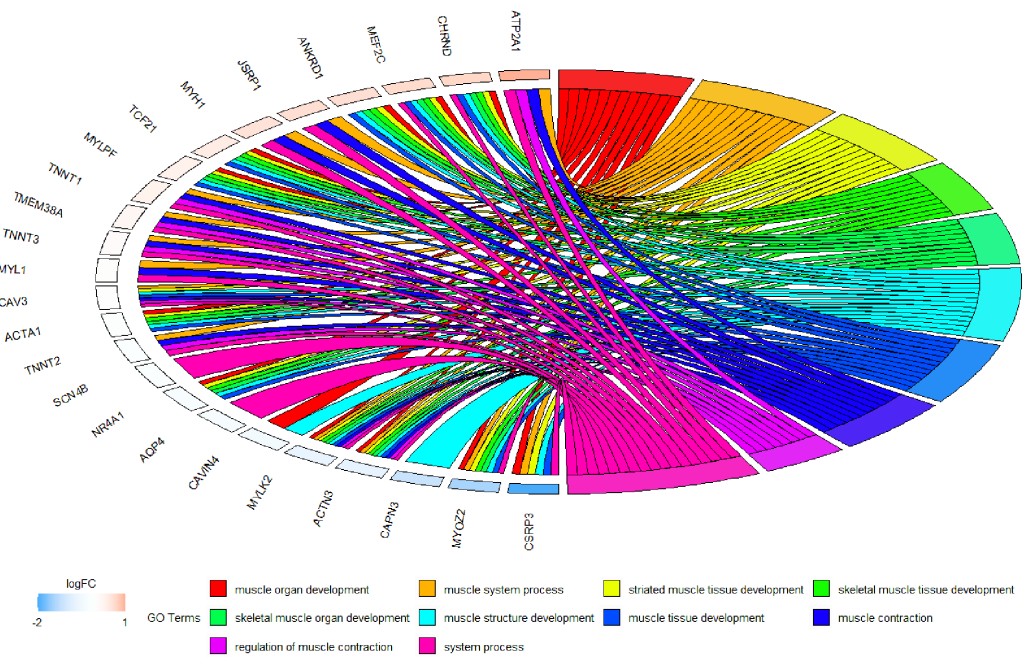

**Figure 13** **Circos plot to indicate the relationship between hub genes and KEGG pathways.**

mice was 1.5 times higher than that of 5-day-old mice (*Yuan, Arnold & Jorgensen, 1991*; *Brillantes, Bezprozvannaya & Marks, 1994*; *Kyselovic et al., 1994*; *Rosemblit et al., 1999*; *Rossi et al., 2007*). Myosin binding protein C (Mybpc) located on the sarcomere mainly includes three subtypes: skeletal slow type (Mybpc1), skeletal fast type (Mybpc2) and cardiac type

(Mybpc3), which can interact with a variety of contractile and structural protein (*Bennett, Furst & Gautel, 1999*; *Winegrad, 1999*). Studies have shown that lack of Mybpc2 expression can lead to severe skeletal myopathy, increased apoptosis, up-regulated expression of factors related to muscle protein degradation, accompanied by structural changes and muscle weakness (*Li et al., 2016*). LncFAM is a new muscle-specific lncRNA, which can recruit RNA-binding protein HNRNPL into the Mybpc2 promoter, thus increasing *Mybpc2* mRNA transcription and myogenic protein Mybpc2 production, and ultimately promoting the differentiation of human myoblasts into myotubes (*Chang et al., 2022*). The above studies suggest that these five hub genes may play an important role in muscle growth and development, and the molecular mechanism needs further study.

## CONCLUSION

We constructed a gene co-expression network using WGCNA, measured the relationship between genes and modules, and explored the relationship between modules and clinical features. We also screened for modules associated with muscle tissue and identified five hub genes, including *Atp2a1*, *Tmod4*, *Lmod3*, *Ryr1* and *Mybpc2*, which are involved in signaling pathways such as regulation of muscle contraction and muscle structure development. In addition, we used qRT-PCR to detect the expression level of hub gene in bovine BSMSCs in different tissues and different culture periods. It was found that the expression of these hub genes was the highest in muscle tissue and highly expressed in induced and differentiated bovine BSMSCs. Therefore, we believe that these five hub genes are closely related to muscle tissue, and the key genes and pathways identified in this study can provide guidance for further exploration of skeletal muscle development and its physiological regulation mechanism.

## ACKNOWLEDGEMENTS

We would like to acknowledge the GEO and STRING databases for their free use.

### Funding

This study was supported by grants from the National Natural Science Foundation of China (U22A20506, 32072720), the Key Research and Talent Introduction Project of Ningxia Hui Autonomous Region (2021NXZD1, 2021BEF01002) and the cultivation project for talents in science and technology innovation of Ningxia Hui Autonomous Region (2020GKLRLX02). The funders had no role in study design, data collection and analysis, decision to publish, or preparation of the manuscript.

### Grant Disclosures

The following grant information was disclosed by the authors:
The National Natural Science Foundation of China: U22A20506, 32072720.
The Key Research and Talent Introduction Project of Ningxia Hui Autonomous Region: 2021NXZD1, 2021BEF01002.

The cultivation project for talents in science and technology innovation of Ningxia Hui Autonomous Region: 2020GKLRLX02.

## Competing Interests

The authors declare there are no competing interests.

## Author Contributions

- Junxing Zhang conceived and designed the experiments, performed the experiments, analyzed the data, prepared figures and/or tables, authored or reviewed drafts of the article, and approved the final draft.
- Hui Sheng conceived and designed the experiments, performed the experiments, analyzed the data, authored or reviewed drafts of the article, and approved the final draft.
- Cuili Pan analyzed the data, authored or reviewed drafts of the article, and approved the final draft.
- Shuzhe Wang analyzed the data, authored or reviewed drafts of the article, and approved the final draft.
- Mengli Yang analyzed the data, prepared figures and/or tables, and approved the final draft.
- Chunli Hu analyzed the data, prepared figures and/or tables, and approved the final draft.
- Dawei Wei analyzed the data, prepared figures and/or tables, and approved the final draft.
- Yachun Wang conceived and designed the experiments, authored or reviewed drafts of the article, and approved the final draft.
- Yun Ma conceived and designed the experiments, authored or reviewed drafts of the article, and approved the final draft.

## Animal Ethics

The following information was supplied relating to ethical approvals (i.e., approving body and any reference numbers):

This study was approved by the Animal Welfare Committee of Ningxia University (permit number NXUC20211058).

## Data Availability

The data is available at NCBI GEO: GSE137943, GSE116775. All the data used in this study are available in the Supplemental Files.

## Supplemental Information

Supplemental information for this article can be found online at http://dx.doi.org/10.7717/peerj.15093#supplemental-information.

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
