# Peer review of "Identification of key genes in bovine muscle development by co-expression analysis"

_PeerJ, doi:10.7717/peerj.15093_

## Round 0.1 · original submission · Major Revisions

The reviewers have highlighted several areas where your manuscript may be improved and further clarification is needed.


·

Basic reporting

This work investigates the differential expression analysis of 5 selected hub genes identified using published datasets that could play a role in muscle development. The authors used bioinformatic tools to reveal the potential biological processes and functions in which these genes can be implicated. They also verified their expression using primary bovine skeletal muscle satellite cells. They conclude that these genes seem to be involved in muscle development.
The authors used a variety of bioinformatic approaches to obtain the results and draw the conclusions. A main issue is the use of the primary cells which needs clarifications on multiple levels. The authors need to address the following points to improve the quality of the manuscript.

Professional English is used throughout. Literature references are appropriate, except when mentioned otherwise in the review report. Sufficient context provided.

Experimental design

1. Line 11, secretion of what?
2. Please use proper gene and protein symbols formatting throughout the manuscript, e.g. in the abstract line 21 replace ATP2A1, TMOD4, LMOD3, RYR1 and MYBPC2 with Atp2a1, Tmod4, Lmod3, Ryr1 and Mybpc2.
3. Lines 59-65, the description of the experimental design is confusing and needs rewriting.
4. Line 61, references of the previous studies must be included.
5. It is not clear if the animals used were from same or different breeds. This needs clarification.
6. Line 105, what do the authors mean by “eventually bovine BSMSCs were isolated”? The exact procedure must be described in detail or reference needs to be provided.
7. Were the cells cultured and differentiated in dishes, flasks or plates? At what density? Essential details are missing.
8. Line 111, which kit?
9. Line 116, washing not cleaning, line 117, blocked not sealed.

Validity of the findings

10. Fig2, what is the brown, blue etc colors? These need to be defined appropriately.
11. What is GSE116775 dataset? This is not mentioned in methods.
12. Lines 178-187 and Fig10. What is the rationale of this experiments and why these results are in the main manuscript? It is well known that MyHC and MyoG are differentiation markers and therefore the results presented here are expected. Immunofluorescence for the translated “hub genes” would be more interesting.
13. The discussion is very general and should be more focused on the possible role of the selected genes identified by this study. It is not clear how these genes can be used in future studies or applications to study muscle development or breeding.

Additional comments

It would be useful for the authors to carry out a similar analysis using different breeds in order to facilitate bovine molecular genetic breeding as they state in the abstract

Reviewer 2 ·

Basic reporting

1. All Figures should have legends, which should explain the basic information of the Figures in detail. Please supplement or add more detailed descriptions to all Figures.
2.Please include MYBPC2 primer information into the Supplementary Table 8.
3.Line 60 “lng” should be “lung”; line 200 “datah” should be “data”.

Experimental design

1. Please describe the calculation method of qRT-PCR and the internal reference genes used in the qRT-PCR experiments.
2.I did not find the cq data of internal reference gene in the qRT-PCR experiment data of ”peerj-79674-qRT-PCR_of_cell_expression_profile” and ”peerj-79674-qRT-PCR_results_of_tissue_expression_profile”, please list it.
3.Line59-60, “we collected tissue samples of kidney, liver, heart, muscle, rumen, adipose, spleen and lng of a premature calf”,do the tissue samples used for qRT-PCR experiment come from this premature calf ? Whether the tissue samples and cells used for qRT-PCR experiment have biological duplication? Please indicate these basic information in the materials and methods or in the legend of Figure 9 and Figure 10.
4.If possible, please indicate the age of the cattle (Line 69-70) used for tissue expression profile analysis.

Validity of the findings

1.Authors used WGCNA and GSEA to study the RNA-seq data sets of bovine tissue differential expression analysis, and 5 core genes were screened, and it was also verified that these five genes were highly expressed in bovine muscle tissue and differentiated BSMSCs. The growth and development of animal muscle is a complex and multi-stage process. The degree of muscle development of newborn calves, especially premature calves, may be quite different from that of adult calves. According to the description in materials and methods, the author seems to use the tissues of premature calves for qRT-PCR verification. If these 5 genes can be used as marker genes or key genes of muscle, then their expression in adult cattle is worth testing, which will be more convincing. If possible, please add the expression data of these genes in muscle tissue of adult cattle.
2.In this study, it is considered that the five selected genes can be used as muscle marker genes or key genes. From this point of view, analyzing the expression of the protein is more indicative of the potential function of the gene than the expression of mRNA. Discovering new muscle marker genes is of great significance for scientific research or gene breeding, so in order to confirm the significance of this study and give better reference to peers, please supplement the experiments on the protein expression level of these core genes.

---

## Round 0.2 · accepted · Accept

Your resubmitted manuscript has been re-assessed by one of the original reviewers. They are satisfied that the changes made to the manuscript have addressed all points raised. At this point, the paper is ready for publication.

·

Basic reporting

The authors addressed my comments to a satisfactory level and the manuscript can be accepted for publication in this revised form.

Experimental design

The authors addressed my comments to a satisfactory level and the manuscript can be accepted for publication in this revised form.

Validity of the findings

The authors addressed my comments to a satisfactory level and the manuscript can be accepted for publication in this revised form.